# Analysis of Hydrogen Distribution and Diffusion in Pre-Strained SUS316L through Scanning Kelvin Probe Force Microscopy and Thermal Desorption Spectroscopy

**Shuanghe Chi [1], Jinxing Guo [2], Zhengli Hua [1,*], Juan Shang [1] and Baihui Xing [1]**

[1] Institute of Process Equipment, Zhejiang University, Hangzhou 310027, China
[2] Xi'an Aerospace Propulsion Institute, Xi'an 710077, China
[*] Correspondence: huazhengli007@126.com

**Abstract:** Austenitic stainless steels ($\gamma$-SS) play an important role in the storage of high-pressure hydrogen. However, hydrogen embrittlement (HE) can significantly degrade the mechanical properties of $\gamma$-SS. Measuring the distribution of hydrogen in $\gamma$-SS is a vital way to learn about HE. In this paper, scanning Kelvin probe force microscopy (SKPFM) and thermal desorption spectroscopy (TDS) have been utilized to analyze the distribution and diffusion of hydrogen in pre-strained SUS316L. Additionally, the McNabb–Foster model is employed to calculate hydrogen in the lattice and phase boundaries along the sample's thickness direction. The results demonstrate that the combination of SKPFM and TDS is an effective approach for studying hydrogen distribution and diffusion in metals. It was observed that hydrogen segregation occurs at the boundary between the martensitic ($\alpha'$) and austenite ($\gamma$) phases. The inhibitory effect of the oxide film on hydrogen diffusion is more significant at lower temperatures. However, it should be noted that the McNabb–Foster model exhibits relatively high accuracy in predicting hydrogen desorption at higher temperatures while disregarding the influence of the native oxide film.

**Keywords:** scanning Kelvin probe force microscopy (SKPFM); SUS 316L; thermal desorption spectroscopy (TDS); hydrogen embrittlement

## 1. Introduction

As the global economy continues to develop rapidly, the issue of energy scarcity is becoming increasingly severe. Moreover, the ecological degradation caused by energy consumption is becoming more apparent. To address these challenges, countries worldwide are actively pursuing energy structural transformation, which involves shifting from fossil fuels to renewable energy sources. Wind and solar energy are highly favored for their clean and pollution-free attributes as well as the well-developed industrial chains supporting them. Because of their intermittent and unpredictable nature, it is difficult to store the electricity generated by large-scale wind and solar power conversion efficiently. As a result, a significant amount of electricity is wasted. To address this issue, the surplus electricity can be transformed into hydrogen for storage and utilization. Hydrogen is a highly promising energy source that is globally recognized for its high calorific value and environmental friendliness [1,2]. Safe, cost-effective, and efficient storage and transportation technologies play a crucial role in the utilization of hydrogen energy. High-pressure gaseous hydrogen storage stands out among various methods due to its superior equipment structure and fast filling speeds. Depending on the scenario in which it is used, it can be classified into three categories: stationary high-pressure hydrogen storage vessels, transportable hydrogen storage pressure vessels, and vehicle-mounted hydrogen cylinders.

Austenitic stainless steel is highly valued for its exceptional properties, including corrosion resistance and ease of processing, making it a popular choice for high-pressure

hydrogen storage systems. Furthermore, austenitic stainless steel exhibits a lower hydrogen diffusion coefficient and greater hydrogen solubility, which reduces the likelihood of hydrogen-induced brittle fracture. As equipment is used for a longer period of time, it becomes increasingly difficult to prevent the occurrence of hydrogen embrittlement (HE), which can significantly degrade the mechanical properties of $\gamma$-SS. This concern about the dependability and security of high-pressure storage hydrogen systems has prompted numerous studies on the mechanical performance of $\gamma$-SS in a hydrogen atmosphere [3,4]. Yamabe et al. [5] investigated the influence of internal and external hydrogen on slow strain rate tensile performance of solution-treated 300 series Austenitic stainless steel at room temperature. They believed that the nickel-equivalent content had an impact on the hydrogen embrittlement resistance of steel. Matsuo et al. [6] conducted a study on the mechanical properties of pre-strained 316L steel under a pressure of 10 MPa and a temperature of 250 °C. Their SEM results showed that it was possible to observe the presence of dimples on the sample surface. Further analysis by the author revealed that the small dimples consisted of interconnected voids formed due to localized shear strain. Qu et al. [7] conducted a study using SSRT and TDS to investigate the impact of plastic deformation on hydrogen diffusion in S30408. The researchers concluded that plastic deformation affected hydrogen diffusion through the control of dislocations and the induction of strain-induced martensite, which is a complex process.

It is worth pointing out that many researchers have identified a relationship between strain-induced martensite $\alpha'$ and HE in $\gamma$-SS. Mine et al. [8] proposed a mechanism in which an excessive amount of hydrogen generated during the transformation of $\gamma$ into $\alpha'$ in a hydrogen atmosphere accumulated within the retained austenite. This accumulation of hydrogen led to localized deformation and subsequent fracture. Bak et al. [9] investigated the impact of strain-induced martensite (SIM) on the mechanical characteristics of AISI 304 in a hydrogen environment by controlling the rate of tension. Their findings indicated that the material's strength and elongation after fracture improved due to SIM being present in the air environment. However, the introduction of hydrogen hindered the formation of SIM, resulting in localized brittle fractures. Zhang et al. [10] discovered that strain-induced $\alpha'$ provided a pathway for hydrogen diffusion, resulting in hydrogen segregation and promoting crack initiation.

Investigating the evolution of hydrogen distribution and diffusion in different microstructures is crucial for understanding the connection between strain-induced $\alpha'$ and HE. However, accurately measuring the distribution of hydrogen in materials poses significant challenges. There are various techniques available for analyzing the distribution of hydrogen in metals. In the initial stages, Hydrogen Microprint (HMT) is widely used. Pu et al. [11] used HMT to explore the transport behavior of hydrogen in metals. Their findings revealed that hydrogen tended to accumulate along slip bands, supporting the idea that dislocations serve as traps for hydrogen. Brandolt et al. [12] utilized the HMT technique to directly observe the trapping sites for hydrogen in P110-containing coatings. Their results demonstrated that hydrogen has a higher tendency to accumulate at interlayer interfaces and within coating defects. However, HMT only provides qualitative information about the hydrogen emitted from the specimen within a specific period and lacks real-time monitoring of hydrogen concentration within the specimen, presenting significant limitations [13,14]. Another technique, Secondary Ion Mass Spectrometry (SIMS), offers quantitative detection of hydrogen distribution in materials. Wang et al. [15] conducted a study to examine the distribution and content of hydrogen in AISI430 using the TOF-SIMS and TDS techniques. Their findings revealed that, in comparison to grain boundaries, carbides possess a higher capacity for trapping hydrogen and serve as irreversible hydrogen traps. Wada et al. [16] employed SSRT and SIMS techniques to investigate the behavior of hydrogen within materials. Their research findings indicated that hydrogen tends to segregate at grain boundaries, leading to an increased likelihood of intergranular fracture. Unfortunately, SIMS is prone to signal interference from the surrounding environment [17,18]. Over the past few years, scanning Kelvin probe force microscopy

(SKPFM) has gained recognition as an effective and non-invasive technique for high-resolution mapping of localized hydrogen distribution [19]. It is worth mentioning that SKPFM enables characterization of fluctuations in the Contact Potential Difference (CPD) resulting from hydrogen invasion. CPD is calculated using the following formula: CPD = $(\varphi_{tip} - \varphi_{sample})/e$, where $\varphi_{tip}$ and $\varphi_{sample}$ represent the work functions of the tip and sample, respectively, and e denotes the elementary charge [20]. According to this formula, the CPD value increases as $\varphi_{sample}$ decreases, while $\varphi_{tip}$ remains constant. Hydrogen has the ability to modify the work functions of a sample. The localized hydrogen content in the material can be determined by measuring the variations in CPD. A number of researchers have utilized SKPFM to analyze hydrogen concentration and segregation in metals. For example, Wang et al. [21] studied the evolving behavior of hydrogen dependence in duplex stainless steel using SKPFM, and Nagashima [22] utilized silver decoration in conjunction with SKPFM to investigate the distribution of hydrogen in dual-phase austenite-martensite steel. Ma et al. [23] utilized SSRT and SKPFM techniques to examine the influence of dislocations on hydrogen behavior in single-crystal nickel. Their research revealed that while dislocations did not speed up hydrogen movement, they could facilitate the uphill transport of hydrogen atoms. However, SKPFM alone cannot provide quantitative information on the hydrogen content in the material.

To address this limitation, thermal desorption spectroscopy (TDS) can be employed. TDS is a non-isothermal technique that generates a curve showing the desorption of hydrogen. The total amount of hydrogen in the sample and the activation energy of hydrogen traps can be determined through calculations [24]. The extent of hydrogen embrittlement damage in ultra-low-carbon steels was evaluated through the utilization of TDS [25]. The TDS outcomes had a strong correlation with the microstructural properties of the material. In TDS, a pre-charged sample undergoes a specific temperature profile, typically characterized by a continuous heating rate. The released hydrogen gas from the specimen is recorded by a mass spectrometer during the heating process. The combination of SKPFM and TDS provides an essential approach for studying the evolution of hydrogen distribution and diffusion within the material.

In this study, SKPFM was employed to investigate the distribution of hydrogen in thermally hydrogen-charged SUS316L containing strain-induced $\alpha'$. Additionally, the amount of hydrogen released from the sample under similar conditions was measured using TDS. By analyzing the changes in CPD and the quantity of released hydrogen, the behavior of hydrogen in the metal was characterized in terms of time, space, and quantity.

## 2. Materials and Methods

Commercially available SUS316L was used, which has a chemical composition of 10.15Ni-16.73Cr-1.17Mn 0.018C-0.56Si-0.033P-0.001S-bal. Fe (units in mass %). The material was converted into cylindrical tensile specimens measuring 5 mm in diameter. Tensile testing was carried out at room temperature with a tensile rate of 0.2 mm/min. Plate-shaped samples measuring 1 mm in thickness and 5 mm in diameter were then obtained from the area where necking was observed in the direction of tension. These specimens were then subjected to SKPFM and TDS analysis, respectively. After grinding with 2000-grade SiC papers, the specimens were electropolished and then cleaned in ethanol using ultrasonic methods before SKPFM and MFM. Subsequently, all the specimens underwent gaseous hydrogen charging in a high-pressure hydrogen charging reactor (at 70 MPa hydrogen pressure and 300 °C temperature for 250 h). To prevent hydrogen gas from escaping, the hydrogen-charged specimens were kept in a refrigerator at −18 °C until measurements commenced. SKPFM combined with Magnetic Force Microscopy (MFM) technology was performed in a flowing nitrogen atmosphere with 99.9995% purity at a temperature of 20 ± 0.5 °C. A Nanoscope IIIa scanning probe microscope outfitted with a custom-designed nitrogen chamber was utilized to mitigate the influence of oxygen and water vapor on the SKPFM results. The probe used in the SKPFM test was MESP, with a first-order resonance frequency of 60~100 kHz. It had a holder length and width of 200~250 μm and 2.2~3.5 μm,

respectively, and was coated with chromium and cobalt for reflective and conductive surfaces. The lift height of the probe was set to 50 nm and the excitation intensity was set to 3 V. The excitation frequency and phase were fine-tuned based on the first-order resonance frequency of the probe to ensure stable measurements of CPD. Other testing parameters in SKPFM were optimized following the methods proposed by Jacobs and Mélin [26,27]. In our experimental conditions, the difference between CPDs measured at different time points (each corresponding to a distinct approach of the SKPFM probe) was within 0.02 V. To enhance the rate of hydrogen desorption in the hydrogen-charged specimen, the sample was heated in a vacuum furnace. The vacuum furnace system was primarily composed of a heating system, a vacuum system (including mechanical and molecular pumps), a cooling system, and the vacuum furnace itself. It was capable of reaching a maximum heating temperature of 1000 °C and a vacuum degree of $1.2 \times 10^{-5}$ Torr. Based on earlier research, we discovered that temperatures below 400 °C did not affect the surface potential of the sample and that the CPD value remained constant. Additionally, if the temperature exceeded 400 °C, carbide precipitation occurred at the grain boundaries, which could lead to hydrogen segregation. The sample underwent two heating–keeping–cooling processes. The first process involved heating the specimen to 150 for 1 h, while the second process involved heating it to 300 °C for 1.5 h. After two heating cycles, almost all of the hydrogen had been desorbed from the sample. During heating, a linear temperature increase was applied. Further details regarding the temperature evolution can be observed in Figure 1.

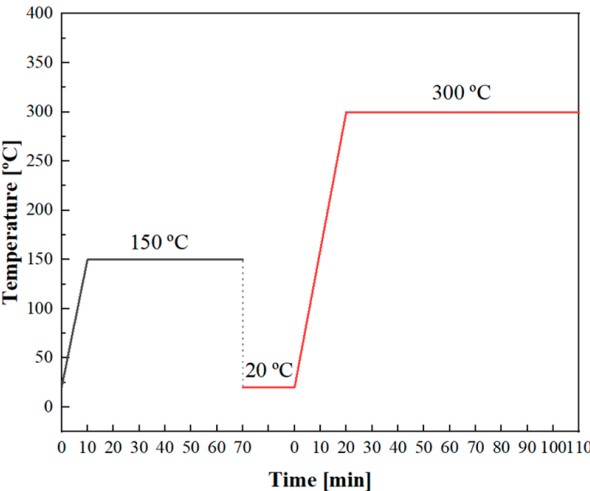

**Figure 1.** Heating process for the specimens within the vacuum furnace.

The TPD Workstation test device, manufactured by the British company Hiden, was utilized in TDS tests. The device had a maximum temperature capability of 1000 °C and could achieve a vacuum level of $10^{-9}$ Torr within the test environment chamber. In TDS tests, hydrogen-charged plate specimens were employed to examine the relationship between hydrogen distribution and its subsequent release. The initial heating rate was 13 °C/min and the duration of heating was 10 min, during which the temperature was raised from 20 °C (room temperature) to 150 °C. The second heating rate was 14 °C/min, and the heating time was 20 min, during which the temperature was increased from 20 °C (room temperature) to 300 °C. The temperature variation process during TDS tests mirrored that of the SKPFM tests.

## 3. Results and Discussion

The images of the sample obtained by MFM and SKPFM were observed prior to hydrogen charging, as depicted in Figure 2. In Figure 2a, the dark region represents $\alpha'$ while the bright region corresponds to $\gamma$ in the MFM image. Figure 2b presents the SKPFM image of the identical region. The nitrogen environment clearly reveals that the CPD of $\alpha'$ is higher compared to $\gamma$, suggesting that the work function of $\gamma$ is greater than that of

$\alpha'$ prior to hydrogen charging. The difference in CPDs in the SKPFM images enables easy differentiation between $\alpha'$ and $\gamma$.

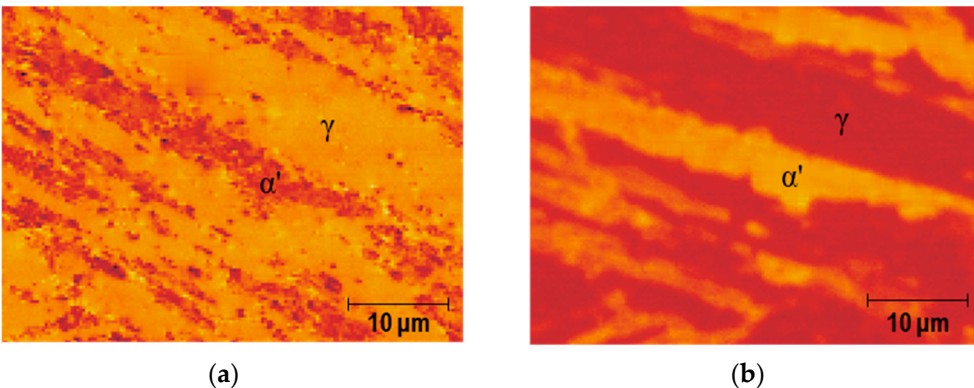

(a)        (b)

**Figure 2.** MFM and SKPFM images in the hydrogen-uncharged specimen: (**a**) MFM image and (**b**) SKPFM image.

After hydrogen charging, the concentration of hydrogen in the $\gamma$ phase is higher than that in the $\alpha'$ phase due to the $\gamma$ phase having a higher solubility for hydrogen. Figure 3 illustrates SKPFM images captured at the same position and different time intervals for the hydrogen-charged specimen. As depicted in Figure 3a,b, following hydrogen charging, the CPD of $\gamma$ surpasses that of $\alpha'$, which is in contrast to the pre-charging scenario. This indicates that hydrogen can reduce the material's work function. The progression of the CPD in the hydrogen-charged specimen along the black line depicted in Figure 3b is depicted in Figure 3e. Over time, hydrogen gradually escapes from the sample surface, leading to a gradual decrease in CPD. At room temperature, the hydrogen desorption rate within the material is slow, leading to considerable retention of hydrogen in the sample. To accelerate hydrogen desorption, the sample underwent heating and was held at 150 °C in a vacuum atmosphere for 1 h followed by rapid cooling to room temperature. In Figure 3c, it can be observed that the difference in CPD between the $\alpha'$ and $\gamma$ phases decreases, with a CPD peak appearing at the interface between $\gamma$ and $\alpha'$. This peak indicates a higher concentration of hydrogen in that specific region. In Figure 3e, a wide decrease in CPD post-heating signifies the escape of a substantial amount of hydrogen from the sample's surface. Subsequently, over a period of 27 h at room temperature, the CPD exhibits an overall increase. The presence of an oxide film inhibits hydrogen release, causing internal hydrogen to diffuse toward the near-surface, resulting in an elevated hydrogen concentration on the metal surface. Following heating at 300 °C, the hydrogen near the surface of the specimen has been released, leading to an increased CPD for $\alpha'$ in comparison to $\gamma$. The CPD resembles its state prior to hydrogen charging. At this stage, only a small amount of hydrogen remains within the material.

While SKPFM can provide qualitative analysis of the evolution of hydrogen distribution on the surface of the specimens over time, it cannot quantitatively measure the emitted hydrogen. To study the relationship between hydrogen release and hydrogen content evolution, TDS was used to analyze the characteristics of hydrogen thermal desorption in materials. To ensure simultaneous hydrogen escape from the metal, consistent temperature variations were maintained, as shown in Figure 1. The relationship between the hydrogen desorption signal intensity (current density) and the elapsed time is depicted in Figure 4a,b. It can be observed that both curves exhibit a similar trend. With increasing temperature, the hydrogen desorption rate gradually increases. During the heat preservation stage, the desorption rate of hydrogen decreases due to concentration differences. Subsequently, the desorption rate of hydrogen further decreases with decreasing temperature. In the TDS curves, the integral of the hydrogen release rate over the elapsed time represents the total amount of hydrogen emitted during the process. In the initial stage, a total of 4.413 weight parts per million (wppm) of hydrogen is released. In the second stage, the total amount of

hydrogen released becomes 40.317 wppm. To acquire the total amount of hydrogen in the material, it was continuously heated up to 800 °C, as depicted in Figure 4c. At this point, there is minimal hydrogen remaining in the material, and the amount of hydrogen released throughout the entire process is 78.053 wppm. Another sample subjected to uninterrupted heating released a total of 75.3236 wppm of hydrogen. The TDS results are largely consistent, indicating that segmented heating has a minimal impact on total hydrogen release. To confirm the accuracy of the TDS test, we performed a calculation using the formula described in [28]. Based on the hydrogen charging conditions in our study, the theoretical hydrogen content was determined to be 87.22 wppm. Accounting for partial hydrogen loss of the material at room temperature we believe that these experimental results are reliable. Through TDS, the amount of hydrogen escaping from samples in SKPFM tests can be determined under different heating conditions.

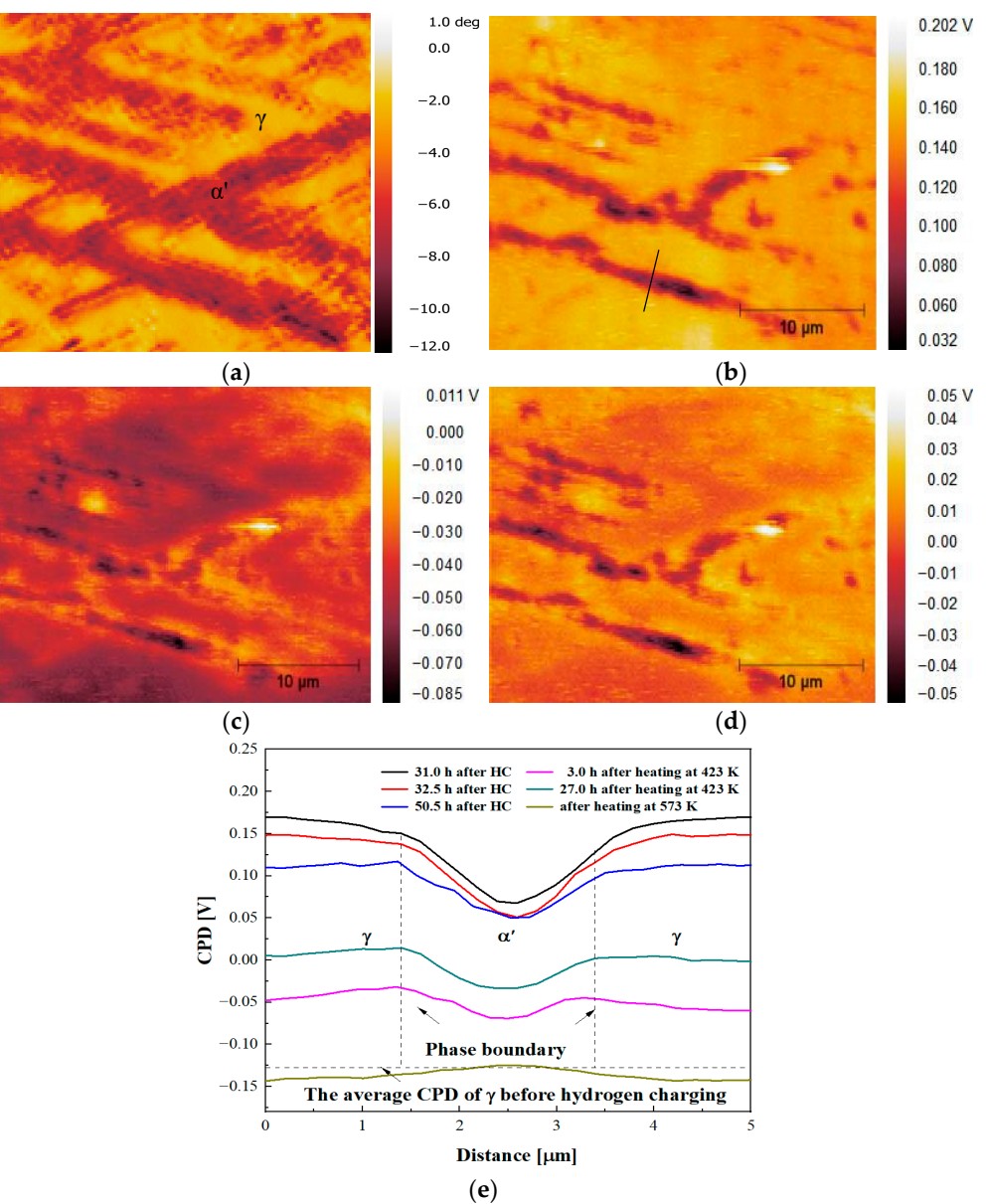

**Figure 3.** (**a**) MFM image; (**b**) SKPFM image captured in nitrogen at t = 32.5 h after sample removal from the refrigerator; (**c,d**) SKPFM images taken in nitrogen after vacuum heating of the sample at 423 K for 3.0 h and 27 h, respectively; (**e**) evolution of CPD in the hydrogen-charged specimen along the dark line indicated in (**b**).

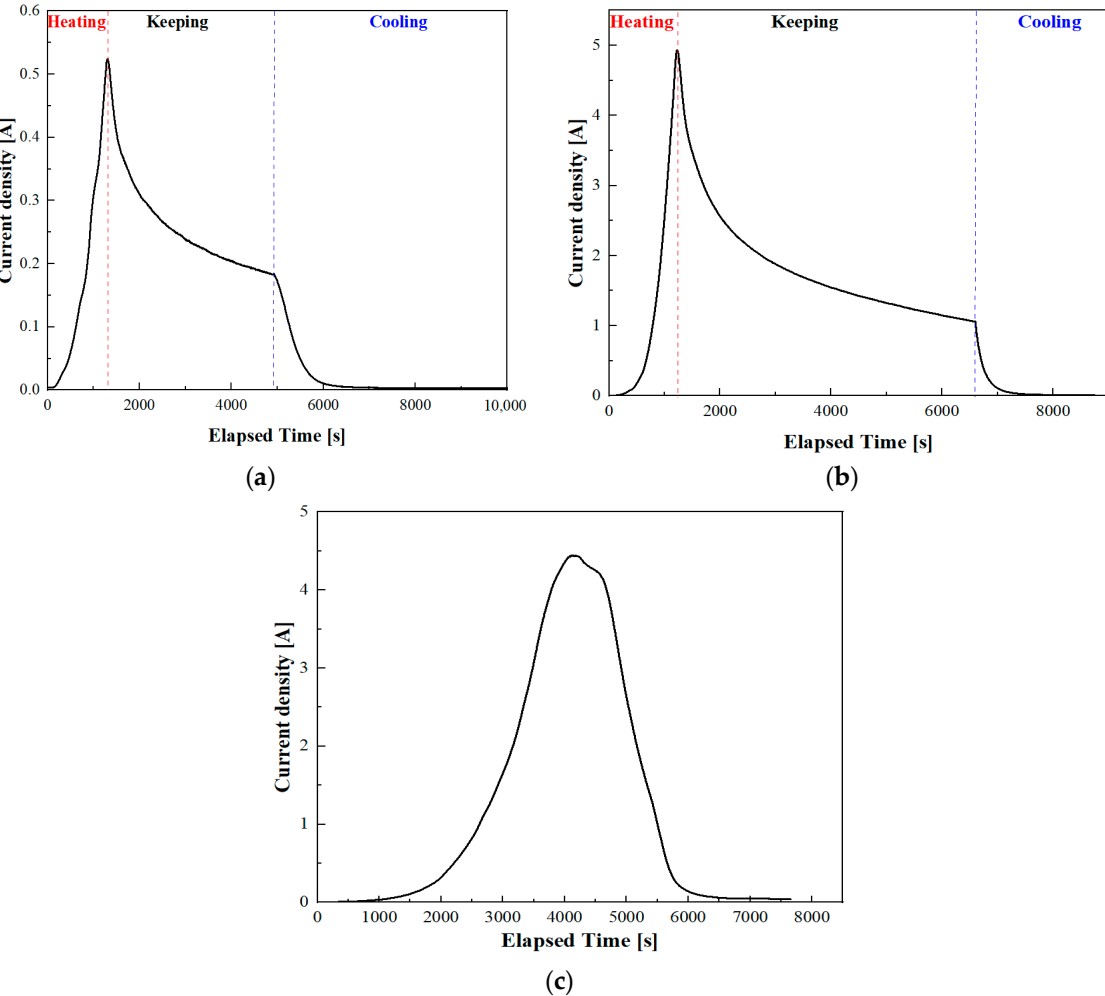

**Figure 4.** Relationship between current density and elapsed time during the TDS test at different temperatures: (**a**) 150 °C, (**b**) 300 °C, and (**c**) 800 °C.

To further analyze the evolution of hydrogen distribution in the material, particularly hydrogen traps such as phase boundaries, we utilized the numerical model proposed by McNabb and Foster [29,30]. This model is utilized to simulate the variation of hydrogen concentration in materials. The diffusion coefficient (D) and solubility (s) of hydrogen in γ were obtained from literature sources. and are provided by [31]:

$$D = 8.9 \times 10^{-7} \times \exp\left(\frac{-53900}{8.314T}\right) \tag{1}$$

$$s = 135 \times \exp\left(\frac{-5900}{8.314T}\right) \tag{2}$$

where T represents the temperature. In the model, we consider the influence of the phase boundary. Previous research has suggested that the trap binding energy of the phase boundary should be set at 45 kJ/mol [32]. Additionally, the assumed trapping site density was $3 \times 10^{-25}$ sites/m³ [33]. According to the condition of gaseous hydrogen charging, we calculated the steady-state hydrogen concentration of the sample using Sievert's law. After hydrogen charging, the boundary condition was set to 0 wppm. Due to the extremely low diffusion coefficient of hydrogen at room temperature, the simulation disregarded the release of hydrogen. Furthermore, we neglected the impact of the oxide film on hydrogen desorption. The simulation results are depicted in Figure 5.

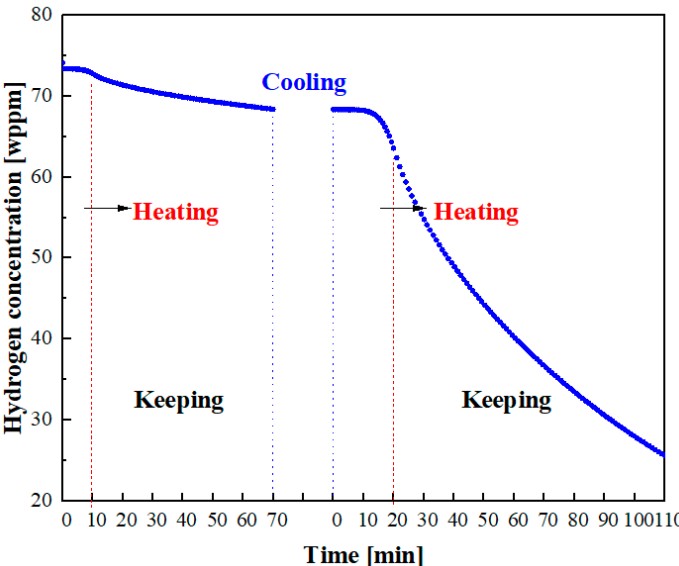

**Figure 5.** Simulated evolution of hydrogen concentration at the near-surface of the sample during the process of heating.

In Figure 5, it can be seen that when the temperature increases there is a slight decrease in the total hydrogen content at the sample's near-surface. After heating, when the sample is placed at room temperature, the amount of released hydrogen is neglected due to the low diffusion coefficient at room temperature. Over time, internal hydrogen diffuses to the surface because of the inhibition of the oxide film, causing an increase in CPD. When heated to 300 °C, the hydrogen content decreases significantly; however, there is hydrogen remaining in the sample. The difference between the simulated and TDS experimental values of hydrogen desorption is 29.84% and 6.16%, respectively, as depicted in Figure 6. Li et al. investigated the characteristics of X80 steel oxide film at different temperatures and obtained the steady-state current and breakthrough time of hydrogen permeation through samples with an oxide film [34]. The authors believed the oxide film to be dense and uniform at low temperatures, providing good resistance to hydrogen permeation. Tanaka et al. [35] proposed that the oxide film on the surface of metals affects the permeation of hydrogen into materials. Therefore, stainless steel used in a hydrogen atmosphere can prevent the permeation of hydrogen into the metal due to its passivated surface. Based on previous research, in this paper we consider the oxide film to be an important factor contributing to the large error observed in tests and simulations at 150 °C. The diffusion rate of hydrogen through an oxide film is temperature-dependent, with higher temperatures resulting in higher diffusion rates and a less pronounced hindering impact of the oxide film. With increasing heating temperature, the hydrogen diffusion rate significantly increases and the effect of the oxide film on hydrogen penetration becomes smaller. At 300 °C, the amount of desorbed hydrogen is accurately predicted by the McNabb–Foster model, and the experimental and simulation results are in good agreement.

Figure 7 depicts the hydrogen distribution across the sample's thickness following the heating process at different temperatures. Despite a significant decrease in hydrogen content near the sample surface, a substantial amount of hydrogen remains within the sample. As depicted in Figure 7a, heating at 150 °C affects the hydrogen concentration in the $\gamma$ lattice up to approximately 20% of the sample thickness, while the phase boundary is largely unaffected by heating except for the near-surface. The excess hydrogen emitted by the lattice accumulates at the phase boundary, leading to higher hydrogen content at the phase boundary compared to the lattice, which is consistent with Figure 3c. Upon cooling the sample to room temperature after heating, the internal hydrogen gradually diffuses towards the near-surface. In combination with the barrier effect of the oxide film, this leads to an increase in hydrogen concentration near the surface and a correspond-

ing brief increase in the CPD, as shown in Figure 3d. Figure 7b shows that heating at 300 °C results in a significant reduction in hydrogen at the phase boundary and the γ lattice within the sample due to the increased hydrogen diffusion coefficient, leading to a substantial decrease in CPD, as shown in Figure 3e. Overall, the simulated evolution of the hydrogen concentration corresponds well to the measured variation in CPD using SKPFM. In future studies, a relationship between CPD and hydrogen content will be established using SKPFM in a vacuum combined with TDS in order to quantify the characteristics of hydrogen distribution.

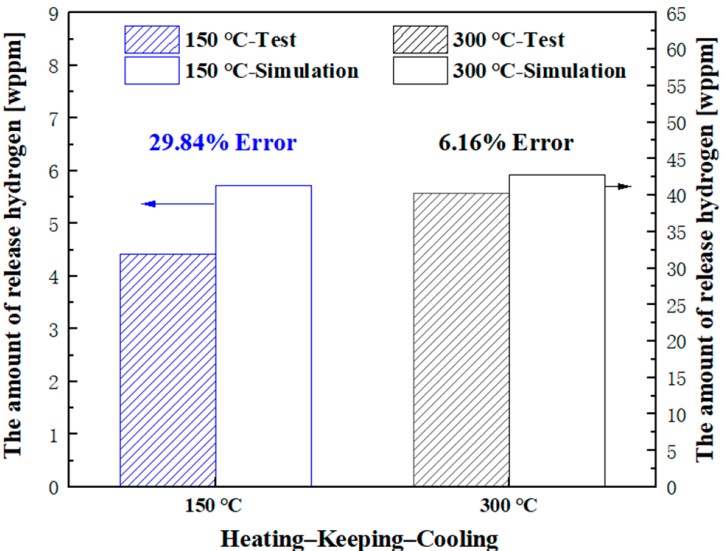

**Figure 6.** Comparison between values obtained from TDS test and simulated values.

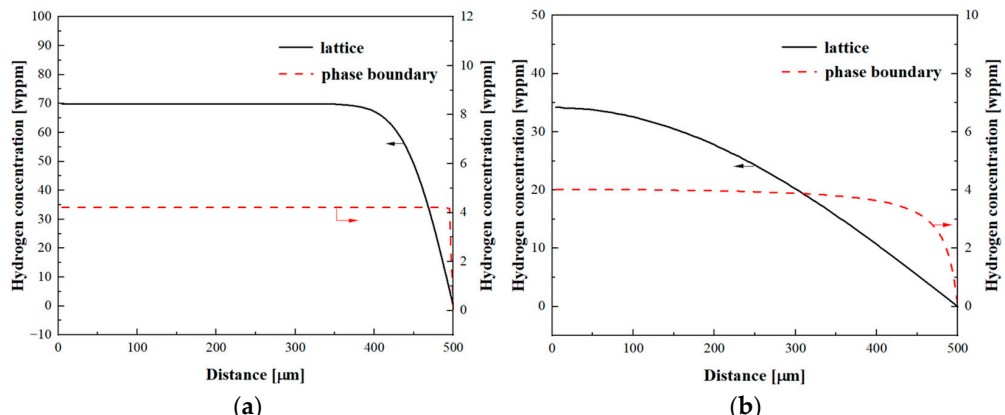

**Figure 7.** Distribution of hydrogen in S31603 along the sample thickness direction after heating: (**a**) 150 °C and (**b**) 300 °C.

## 4. Conclusions

The distribution and concentration of hydrogen in pre−strained SUS316L have been investigated using SKPFM and TDS in combination with numerical simulation. The findings indicate that hydrogen segregation occurs at $\alpha'$ or at the phase boundary between $\alpha'$ and γ due to disparities in hydrogen solubility and diffusion coefficient. A substantial reduction in hydrogen concentration is observed upon heating. Based on the TDS and numerical simulation outcomes, it is observed that the existence of an oxide film at low temperatures hinders the escape of hydrogen from the specimen. However, its impact becomes negligible at high temperatures. The McNabb–Foster model demonstrates relatively high accuracy in predicting hydrogen desorption at elevated temperatures, disregarding the influence of the native oxide film. The simulated variation in hydrogen concentration near the material

surface aligns with the CPD evolution as measured by SKPFM. The combination of SKPFM and TDS effectively captures the behavior of hydrogen within the material.

**Author Contributions:** Conceptualization, S.C. and Z.H.; methodology, J.S.; software, Z.H.; validation, J.S., B.X. and S.C.; formal analysis, J.G.; investigation, J.G.; resources, Z.H.; data curation, J.S.; writing—original draft preparation, S.C.; writing—review and editing, S.C. and Z.H.; visualization, B.X.; supervision, Z.H.; project administration, Z.H.; funding acquisition, Z.H. All authors have read and agreed to the published version of the manuscript.

**Funding:** This research was funded by "Pioneer" and "Leading Goose" R&D Program of Zhejiang (2023C01225), Zhejiang University K.P. Chao's High Technology Development Foundation, and the State Key Laboratory of Clean Energy Utilization.

**Data Availability Statement:** Not applicable.

**Conflicts of Interest:** The authors declare no conflict of interest.

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
