# Peer review of "Analysis of Hydrogen Distribution and Diffusion in Pre-Strained SUS316L through Scanning Kelvin Probe Force Microscopy and Thermal Desorption Spectroscopy"

_energies, doi:10.3390/en16207126_

Round 1

Reviewer 1 Report

1. Please add in introduction information what are the other high pressure hydrogen storage systems.

2. Why authors state that “austenitic stainless steels (γ-SS) are commonly employed” (line 38)  – please put some reference.

3. Why authors choose temperatures of 150 oC and 300 oC? Please put this information in article.

Minor editing errors:

1. Please put capital letter at the beginning of sentence – line 89, and correct this sentence.

2. Please add in capture figure 2 information “martensitic (α') and austenite (γ) phases”

3. Please put the units in graphs in brackets, for example: Time [min], Distance [µm]

4. Please add information in line 180 wppm (Weight Parts per Million)

5. Please put equations( line 199, 200)  in single lines and please number them

Reviewer 2 Report

Dear authors, the solubility of hydrogen in stainless steels has been measured to be lower than 40ppm (https://doi.org/10.1016/j.ijhydene.2006.05.008), yet your study claims to detect closer to 80ppm without explanation or enough experimental details explaining how this number was achieved and why it is so high. 

This paper needs much more details pertaining to experimental details such as technique descriptions, measurements devices, measurement methods, etc. 

No comments, minor revisions needed.

Round 2

Reviewer 2 Report

Dear authors, the added information makes the paper clearer. 

Thank you for the additional details. 

Some typos and grammar
